



# Position correction in dust storm forecast using LOTOS-EUROS v2.1: grid distorted data assimilation v1.0

Jianbing Jin[1,2], Arjo Segers[3], Hai Xiang Lin[2], Bas Henzing[3], Xiaohui Wang[2], Arnold Heemink[2], and Hong Liao[1]

[1]Jiangsu Key Laboratory of Atmospheric Environment Monitoring and Pollution Control, Collaborative Innovation Center of Atmospheric Environment and Equipment Technology, School of Environmental Science and Engineering, Nanjing University of Information Science and Technology, Nanjing, China
[2]Delft Institute of Applied Mathematics, Delft University of Technology, Delft, the Netherlands
[3]TNO, Department of Climate, Air and Sustainability, Utrecht, the Netherlands

**Correspondence:** Jianbing Jin (jianbing.jin@nuist.edu.cn) and Hong Liao (hongliao@nuist.edu.cn)

**Abstract.** When calibrating simulations of dust clouds, both the intensity and the position are important. Intensity errors arise mainly from uncertain emission and sedimentation strengths, while position errors are attributed either to imperfect emission timing, or to uncertainties in the transport. Though many studies have been conducted on the calibration or correction of dust simulations, most of these focus on intensity solely, and leave the position errors mainly unchanged. In this paper, a grid dis-
5 torted data assimilation, which consists of an imaging morphing method and an ensemble-based variational assimilation, is designed for re-aligning a simulated dust plume to correct the position error. This new developed grid distorted data assimilation has been applied to a dust storm event in May 2017 over East Asia. Results have been compared for three configurations: a traditional assimilation that focuses solely on intensity correction, a grid distorted data assimilation that focuses on position correction only, and the hybrid assimilation that combines these two. For the evaluated case, the position misfit in the simu-
10 lations is shown to be dominant in the results. The traditional emission inversion improves only slightly the dust simulation, while the grid distorted data assimilation effectively improves the dust simulation and forecast. The hybrid assimilation that corrects both position and intensity of the dust load provides the best initial condition for forecast of dust concentrations.

## 1 Introduction

Dust storms are a result of wind erosion liberating particles from exposed dry surfaces (World Meteorology Organization,
2019). They occurre commonly in arid or semi-arid regions, e.g., North Africa, the Middle East, Southwest Asia and East Asia (Shao et al., 2013). During dust events, fine dust particles can be lifted several kilometers high into the atmosphere, and carried over thousands of kilometers (Zhang et al., 2018). It is estimated that 2000 Mt dust is emitted into the atmosphere annually (Shao et al., 2011). Such huge amount of atmospheric mineral dust has profound effects on the Earth system, e.g., the cycles of energy, carbon and water (Mahowald et al., 2010). Specifically, dust particles are recognized in fertilizing terrestrial
and ocean ecosystem (Shepherd et al., 2016), enhancing precipitation by acting as droplet nuclei (Benedetti et al., 2014), interacting with atmospheric radiation, and may therefore significantly modify the Earth radiative balance (Balkanski et al.,





2007; Wu et al., 2016). Apart from the influence on the environment, dust storms pose a great threat on the human health by carrying thousands tons of particulate matter as well as bacteria, viruses and persistent organic pollutants to densely populated regions (World Meteorological Organization, 2017; Basart et al., 2019). Reported illnesses include dust pneumonia, strep throat, cardiovascular disorders and eye sicknesses (Shao and Dong, 2006; Ozer et al., 2007; Benedetti et al., 2014; World

Meteorological Organization, 2018). The low visibility caused by dusts can also lead to severe disruptions of air and other traffic. For example, more than 1,100 flights were delayed/canceled in Beijing after it was struck by an extreme dust event in May 2017.

Together with growing interest in dust storms, the understanding of the physical processes associated with dust storms has increased rapidly over the last decades (World Meteorological Organization, 2018). Large efforts have been made to develop

dust modeling systems (Marticorena and Bergametti, 1995; Shao et al., 1996; Marticorena et al., 1997; Alfaro et al., 1997; Wang et al., 2000; Liu et al., 2003; Basart et al., 2012), which mathematically simulate the life cycle of dust including emission, transport and deposition. Large scale global dust transport models, e.g., CAMS-ECMWF (Morcrette et al., 2009), or regional ones, e.g., NASA-GEOS-5 (Colarco et al., 2010) and BSC-DREAM8b (Mona et al., 2014), are essential parts of larger Earth system models. The most important application of these models is to forecast dust concentrations over a few hours to a few

15 days in order to reduce the potential threats on society. Though these systems are usually able to predict the starting and ending of a dust event, large differences are found in emission and deposition burdens and spatial distribution of dust clouds (Huneeus et al., 2011; Koffi et al., 2012). Dust simulations could differ from observations up to two orders of magnitudes (Uno et al., 2006; Gong and Zhang, 2008). The modeling skills are limited due to several aspects, e.g., the insufficient knowledge of aerosol size distribution (Mokhtari et al., 2012), mismatch in aerosol removal (Croft et al., 2012), and in particular to

20 the inaccurate quantification of erosive dust emission (Gong and Zhang, 2008; Ginoux et al., 2012; Escribano et al., 2016; Di Tomaso et al., 2017). In addition, the quality of the meteorological data, e.g. wind fields and soil moisture, might strongly impact the prognostic quality of dust emission and transport.

In addition to the efforts of upgrading the physical descriptions in numerical models, data assimilation techniques have been developed to improve simulation of dust loads. Data assimilation aims here to estimate the state of dust concentrations by

25 combining a dynamical model with available observations. An assimilation system could for example adjust model parameters within an allowed range such that a simulation is in better agreement with the observations. Various types of observations have been used to adjust dust simulations, for example particular matter (PM) measurements (Lin et al., 2008; Wang et al., 2008) and visibility records (Niu et al., 2008; Gong and Zhang, 2008) from ground-based monitoring networks, aerosol optical depth (AOD) from sun photometers in the global Aerosol Robotic Network (AERONET) (Schutgens et al., 2012), as well as the

30 satellite retrieved AOD (Khade et al., 2013; Yumimoto et al., 2016; Di Tomaso et al., 2017; Dai et al., 2019). Those studies either focused on updating atmospheric dust concentrations directly, or on optimizing emission parameters that lead to better simulations. In both cases, only the intensity of either concentrations or emissions is adjusted, while other input parameters are assumed to be known and processes of transport and removal are assumed to be certain.

In our previous studies, ground-based $PM_{10}$ (total particulate matter with diameter less then 10 $\mu$m) measurements (Jin et al.,

2018, 2019a) and geostationary satellite AOD (Jin et al., 2019b, 2020) were assimilated with the LOTOS-EUROS simulation





model for dust storm forecasts over East Asia. Also these studies soleley focused on correcting emission intensities. Data selection (Jin et al., 2019b) and observation bias correction (Jin et al., 2019a) were important aspects here to ensure that the available measurements were used correctly. In addition, an adjoint method was used to identify potential new dust emission sources in case the empirical dust emission and its uncertainty scheme cannot fully resolve the observation (Jin et al., 2020).

Severe dust storm events in May 2017 over East Asia were used as test cases, and the assimilation procedure was shown to improve the simulated dust concentrations at time of observation, but also to improve forecasts of dust levels over windows of up to 24 hours. During these studies it was noted that although the modeling system in general provided an accurate forecast of the dust plume, a severe position error was present when the plume traveled over a large distance. Specifically, forecasts by the model simulation reported the dust arrival and departure 1 to 10 hours prior to reality, as will also be illustrated in Section 3.

Position errors are a common problem in meteorology, for example in forecasting hurricanes, thunderstorms, precipitation (Ravela et al., 2007; Nehrkorn et al., 2014, 2015), or meteorology governing events like wildfires (Beezley and Mandel, 2008). In geophysical disciplines, a positional error is often considered together with intensity errors to explain differences between two estimates (Nehrkorn et al., 2015). A misfit in position usually leads to significant degradation of forecasts (Jones and Macpherson, 1997).

When discussing the accuracy of a dust forecast, the shape and position of the plume is a key element, as well as the intensity. The position forecast determines which locations will be affected, when the storm will arrive, and for how long it will last, while the intensity only describes the actual dust level. A dust forecast with position misfit directly results in incorrect timing profiles of dust loads. The information about dust arrival and departure is sometimes more important than the magnitude of dust load in the early warning system, but until now it has attracted only little attention. Facing the unresolved positional mismatch,

the aforementioned data assimilation focusing solely on intensity correction is less effective, as will be illustrated in Section 4.1.

Similar as intensity feature misfits, positional misfits in model simulations can also be adjusted to better resemble observations using data assimilation techniques. Dust simulations suffer from position errors due to for example incorrect emission timing profiles or uncertainties in the transport, both driven by uncertain meteorology fields. To be able to use data assimila-

tion techniques for position correction, it is essential to have a description of these uncertainties. However, position errors are much likely to be non-Gaussian, and not easily captured by a static error covariance model (Nehrkorn et al., 2015). For dust simulation, position errors could be caused by uncertainties in the transport, in particular the wind field. These uncertainties accumulate during the time period from emission in remote desert areas to arrival at observation networks in downwind populated area. Position discrepancies might also arise from incorrect timing profiles of emissions, which is not the case for our

test event as will be explained in Section 3.1. However, determining the covariance either for transport or for emission timing profile is difficult. Even if there is a complex covariance model that could account for the accumulation of uncertainties along the long track of the plume, a substantial amount of observations would then be required to constrain the optimal transport pattern. Data assimilation methods based on static covariance models are therefore often not suitable for dealing with position errors.



Instead, techniques from the field of image processing could be combined with data assimilation to avoid the need for a static covariance that describes the origin of the position error. This has been described as phase-correcting data assimilation in numerical weather prediction (Brewster, 2003), image morphing EnKF for wildfire models (Beezley and Mandel, 2008), grid distortion data assimilation on oil reservoir modeling (Lawniczak, 2012), and in general as position error correction in

variational data assimilation (Nehrkorn et al., 2015). The common approach in all these applications is to re-position the simulation using an image morphing technique, where the optimal morphing parameters are adjusted to obtain the best fit with the observations using data assimilation techniques. In an application with dust plume simulations, the use of image morphing in the data assimilation avoids the need for developing a complex covariance model to describe uncertain transport or emission timing.

In this study, we propose a grid distorted data assimilation method to correct position misfits in a simulated dust plume, which is a novel approach in the context of atmospheric dust modeling. The implemented method offers an efficient way to correct for a phase misfit between a dust simulation and available observations, without changing the transport scheme and/or the emission timing profile. The grid distorted data assimilation is then combined with the emission intensity inversion described in (Jin et al., 2019b) for a *hybrid* method. The *hybrid* method is capable of optimizing the dust plume in case that both

position and intensity misfits are presented in a dust simulation. Starting from the initial condition using the *hybrid* assimilation posterior, dust forecast accuracy (in terms of both arrival and departure, and in actual dust load) is further ensured.

The paper is organized as follows. Section 2 introduces the simulation model and observations used to represent the dust intensity. Section 3 shows an example of a dust position error in a dust simulation. The error source is explained and identified to be the uncertainty in long-distance transport process, and it is illustrated that this uncertainty cannot be explained from the

20 known spread in meteorological forecasts. In Section 4, the necessities of position error correction is emphasized first, and then the methodology of grid distorted data assimilation is introduced. A *hybrid* assimilation method is designed by combining the grid distorted data assimilation and emission inversion in Section 5. The new method is evaluated against assimilation focusing solely on emission intensities or position correction. Section 6 summarizes the conclusion and the added value of using grid distorted data assimilation to resolve model position error.

## 25 2 Dust model and observations

### 2.1 Simulation model

In this study, the dust storm is simulated using a regional chemical transport model, LOTOS-EUROS v2.1 (Manders et al., 2017). LOTOS-EUROS has been used for a wide range of applications supporting scientific research and operational air quality forecasts both inside and outside Europe. At present, the operational forecasts over China are released via the MarcoPolo-Panda

projects (Timmermans et al., 2017; Brasseur et al., 2019) through http://www.marcopolo-panda.eu/forecast/ (last access: July 2020). Besides, it is also implemented in the WMO Sand and Dust Storm Warning Advisory and Assessment System to provide short-time forecast of the dust load over the North Africa-Middle East-Europe areas, the online forecast product are delivered via http://sds-was.aemet.es/forecast-products/dust-forecasts/compared-dust-forecasts (last access: July 2020).





To establish a dust simulation over East Asia, the model is configured on a domain from 15°N to 50°N and 70°E to 140°E, with a resolution about 0.25°× 0.25°. Vertically, the model consists of 8 layers with a top at 10 km. The dust simulation is driven by European Center for Medium-Ranged Weather Forecast (ECMWF) operational forecasts over 3-12 hours, retrieved at a regular longitude/latitude grid resolution of about 7 km. Physical processes included are wind-blown dust emission, diffusion,
advection, dry and wet deposition, and sedimentation.

## 2.2    Observation network

The observations used in this study consist of hourly $PM_{10}$ concentrations from the China Ministry of Environmental Protection (MEP) air quality monitoring network, which is shown in Fig. 1. By now, the network has over 1700 stations, and hence offers an opportunity to track the whole dust plume while it moves through the region.

All these $PM_{10}$ measurements are actually a sum of dust and airborne particles (black carbon, sulphate, etc). Since the analyzed event is an extremely severe case, these $PM_{10}$ measurements were directly used to quantify the dust load in Jin et al. (2019b, 2020). In this study however, an observational bias correction is performed to make the $PM_{10}$ measurements fully representative for the dust loads. First, *non-dust* aerosol levels are calculated using a LOTOS-EUROS simulation following the MarcoPolo-Panda configuration, but with the dust tracers disabled. Using these simulations, bias-corrected dust observations were calculated by subtracting the *non-dust* loads from the original $PM_{10}$ observations. The original $PM_{10}$ measurements vs. the pure dust observations can be seen in Fig. 2(a.1~a.2) and Fig. 7(a.1~a.2). As dust aerosols are far dominant during the severe dust storm, the bias-corrected dust observations are actually very close to the original $PM_{10}$ measurements.

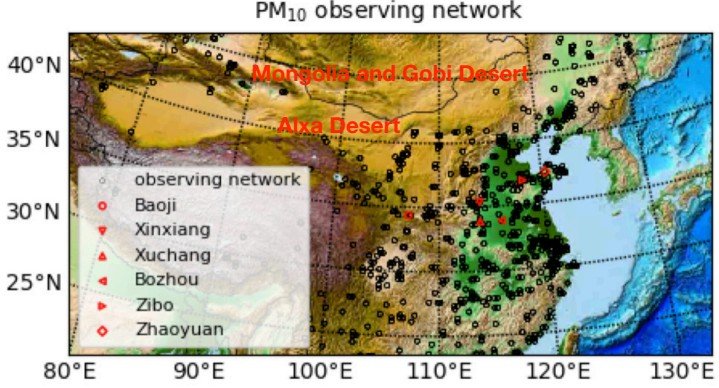

**Figure 1.** East Asia topography map and distribution of the China MEP observing network

## 3    Position error

Numerical dust models are expected to provide correct timing profiles and intensity of dust loads. However, a discrepancy
between observations and simulations is relatively common in terms of both position and intensity. Unlike the intensity esti-





mation that has been widely investigated already, the position error has received less attention, but it has been the main focus of this study.

## 3.1 Position error in dust simulation

The test case investigated in this study is a severe dust storm event that occurred over East Asia in May 2017. The detailed
calibration of the model simulations on this test case can be found in Jin et al. (2019b, 2020). The dust emission occurred from May 02 at the Mongolia, Gobi and Alex deserts, of which the location can be seen in Fig. 1. The dust particles lifted up from these regions were then transported in southeast direction. After 2 to 3 days of transport, the dust plume arrived in central China, where according to the surface observations a positional error was present in the simulations.

The position error in the simulation is illustrated in Fig. 2, which shows the original $PM_{10}$ measurements, bias-corrected dust
observations as well as the *a priori* surface dust concentration (SDC) simulation at May 05 15:00 (China Standard Time, CST). The measurements of $PM_{10}$ are strongly elevated when the dust plume passes, and could increase to values over 2000 $\mu g/m^3$. Under normal conditions the observations (non-dust aerosols) usually do not exceed values of 200 $\mu g/m^3$, and therefore the location of a dust plume is clearly visible in the bias-corrected dust observations, as well as in these original $PM_{10}$ observations. According to the observations in panel (a), the dust plume forms a band from the west to the east over central China. The
corresponding simulation in panel (b) shows a plume with a similar shape, but at a location further to the southeast. This is indicated by the markers that are added to the plumes. For the observations the markers for the left part of the plume are around 35°N and the right one stays around 37.5 °N, while for the simulation they are around 32.5°N and 36°N. The dust plume is therefore positioned about 200 km too far to the south; with a wind speed of 40 km/hour this implies a difference in arrival time of 5 hours. The simulated plume, in particular the left part, is also broader than the rather sharp band that is seen in the
observations.

To quantify the simulation-minus-observation mismatch, the root-mean-square error (RMSE) between dust simulation and bias-corrected dust observation has been computed over all stations in central China (marked by the black framework in Fig. 2a). The RMSE of the *a priori* dust simulation is as high as 388.1 $\mu g/m3$. This vast mismatch is attributed to the sum of intensity and position error (mainly) as will be explained in Section. 5.2.

## 3.2 Uncertainty in emission timing profile

One potential origin of the position error is an incorrect emission time profile. That is, changes in the time period over which dust is released from the source regions could to some extent alter the position of the simulated plume.

Actually during the first 48 hours after dust emission started, the simulated dust plume was still in north China and showed in general the same pattern as visible in the observations. For example the aerosol optical depth (AOD) retrieved from the
Himawari-8 geostationary satellite showed that the simulated plumes are correctly positioned in north China (Jin et al., 2019b). The good phase match in general can also be seen from a snapshot of the ground $PM_{10}$ observation vs. the simulated surface dust concentration at May 04 15:00 (CST) in Fig. 3. There might already be position misfits in the dust simulation at these snapshots, but not easily detected. The magnitudes of the dust concentration showed discrepancies, but these could be corrected



**Figure 2.** Original PM$_{10}$ (a.1), bias-corrected dust observations (a.2); the *a priori* (b), maximum over the ensemble simulations driven by ensemble meteorology (c); posterior dust simulation of the *emis inversion* (d); *grid distorted assim* posterior (e) and *hybrid assim* posterior simulation (f) at 15:00 May 05. SDC: surface dust concentration. Definitions of *emis inversion*, *grid distorted assim* and *hybrid assim* can be found in Table. 1.





by *emission inversion* through assimilating those AOD observations or $PM_{10}$ measurements. The good match in position between simulated and observed dust plume indicates that the emission timing profile is rather accurate too. When the dust plume is transported further southward, the simulated plume starts to deviate from the available surface measurements.

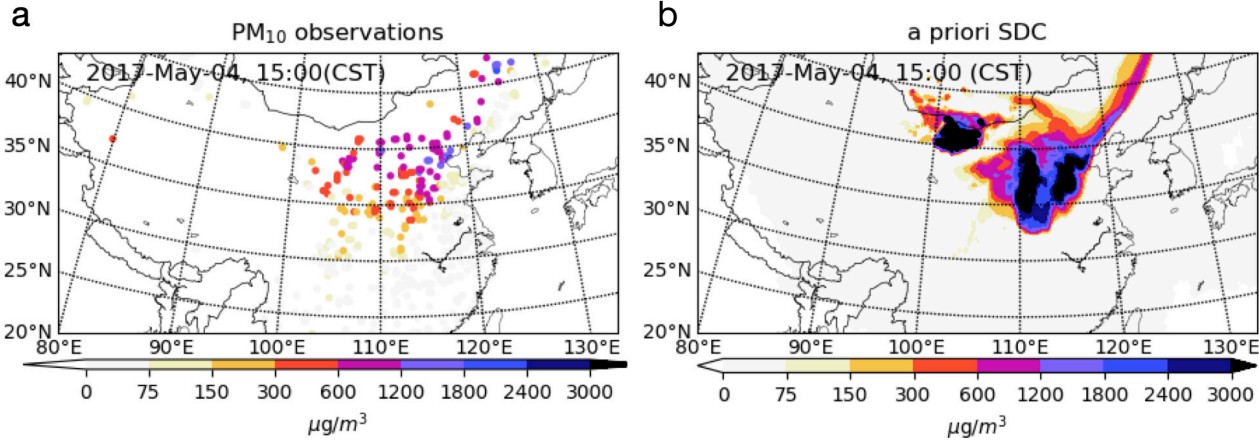

**Figure 3.** $PM_{10}$ observations (a) and the *a priori* dust simulation (b) at 15:00 May 04. SDC: surface dust concentration.

### 3.3 Uncertainty in meteorology

Another possible origin of the position error in the simulations is the uncertainty in the meteorological data. In our study, the simulation model is driven by ECMWF meteorological forecasts. The uncertainty in this input is reflected in the *ensemble forecasts* that are available too (Palmer, 2019). For the studied period, the ensemble forecast of $N_{meteo}$=26 different members is available, where each member is a perturbation of the deterministic forecast.

To estimate the impact of the meteorological uncertainty, the dust simulations have been repeated $N_{meteo}$ times using input
from the meteorological ensemble. The spread in simulated dust concentrations is computed in terms of the maximum over the ensemble via:

$$c_{max}(x,y,z,t) = \max(\, c_1(x,y,z,t), \; \ldots, \; c_{N\,meteo}(x,y,z,t)\,) \tag{1}$$

In here, $c_i$ represents the dust concentration field that results from a simulation with the $i^{th}$ ensemble member. This measure reflects for each location whether in any of the simulations a severe dust load is present. A snapshot of the ensemble maximum
Eq. (1) at 15:00 is shown in Fig. 2 (c). The map shows a broader plume, which implies that some ensemble members result in a dust plume that is more to the north and others more to the south than the *a priori* forecast. The extended dust field is however not wide enough to cover the area with increased observation values. The uncertainty in the meteorological data therefore could not be used to fully account for the position error, and the assimilation system has to correct for this in a different way.





## 4   Grid distorted data assimilation

The experiments in the previous section showed that the mismatch between dust plume simulation and observations cannot be easily explained by inaccurate emission timing or uncertainty in the meteorological data available. We therefore propose to use a *grid distorted data assimilation* to correct for the position errors, without attributing this error to a specific part of the simulation model or its input.

### 4.1   Necessity of position error correction

Position errors pose a great challenge for data assimilation, where it is often easier to adjust amplitudes rather than a position. This strongly limits the forecast skill, and further improvement requires the correction of position errors.

The difference between assimilation of observations with or without correction of position errors is illustrated in Fig. 4. The panels show a hypothetical dust concentrations along a coordinate, which could be either spatial or temporal without loss of generality. The *a priori* simulation (dashed) differs from the observations (stars) both in amplitude and shape (location and width in space, or arrival and duration in time). The underlying simulation model is therefore likely to be imperfect in either emission strengths, emission timing, or transport, or a combination of all of these.

The left panel illustrates a typical assimilation of observed concentrations that adjust emission strengths only. In such an assimilation, the *a priori* concentrations are just scaled towards the observations. The *posterior* concentrations are therefore closer to the observations, but only where the *a priori* simulations has any concentrations at all. At the left side of the axis the simulated concentrations are therefore strongly reduced to match with the zero observations. However, if initially no dust is present in the simulations, as is the case at the right side of the axis, then the assimilation does not suddenly introduce dust out of nothing.

The right panel illustrates how a position error correction could improve this. Before analyzing the observations, the *a priori* plume is shifted and reshaped to have the best match with the observations, ignoring differences in amplitude. If this re-positioned plume is analyzed with the available observations, the *posterior* result is in much better agreement with the observations along the entire axis, also where initially no dust was simulated. The assimilation will still adjust the emission strengths, but these are now not adjusted to correct for transport errors.

### 4.2   Grid distortion

To align the dust plume with the observations, a *grid distortion* method as described by Lawniczak (2012) is used. The procedure is illustrated in Fig. 5. In transport models, the flow equations are usually solved on a discrete grid. For the LOTOS-EUROS model used here, the grid is Cartesian (perpendicular in longitude and latitude), and regular in spacing (panel (a) of the figure). Computed concentrations represent an average over a grid cell, and the simulated plume therefore consists of a set of grid cells with a substantial dust load. Panel (b) shows an example with a dust plume as a band from left to right. The grid distortion smoothly transforms the Cartesian grid into a non-Cartesian grid. That is, the corners of the grid cells are re-positioned to a nearby location, such that each distorted grid cell remains connected to its original neighbors (panel c). The dust concentra-





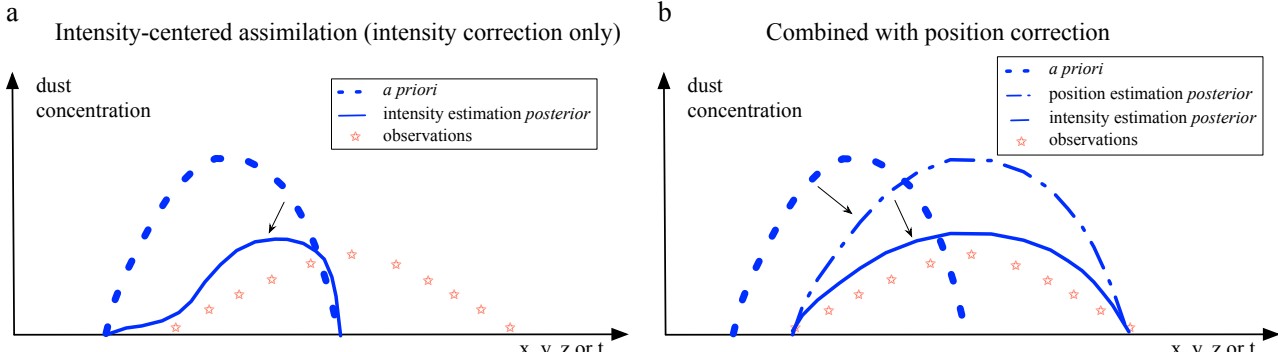

**Figure 4.** Illustration of intensity-centered assimilation only (left) versus assimilation after position error correction (right).

tion in each grid cell in terms of $\mu$g/m$^3$ is kept constant after distortion to ensure a smooth variation of dust intensities over neighboring cells. The dust plume is deformed together with grid (panel d).

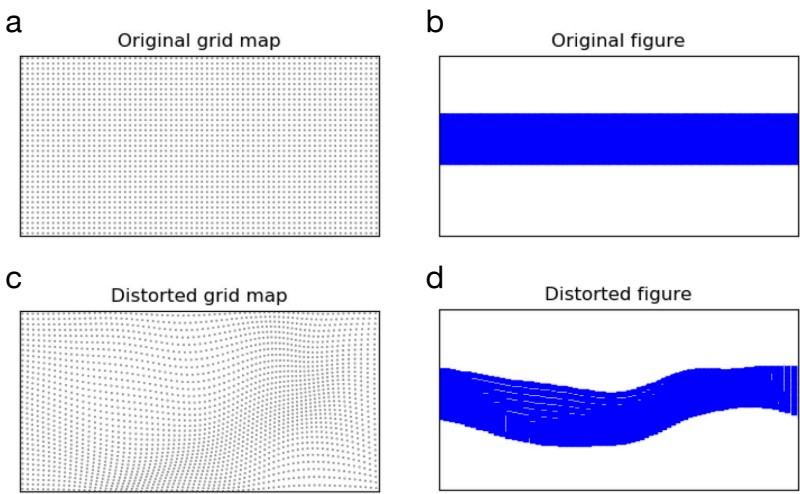

**Figure 5.** Illustration of grid distortion technique: (a) original grid map; (b) original dust concentrations as band; (c) distorted grid map; (d) distorted dust concentrations.

In mathematical formulation, let $(x, y)$ denote the original Cartesian coordinates. A discrete model grid with regular spacing $\Delta x \times \Delta y$ is defined on points $(x_i, y_j)$, with $i$ and $j$ the integer indices of the grid points in $x$ and $y$ direction. The grid distortion

5  is defined as a coordinate transformation that projects an original location $(x, y)$ onto a new location $(\lambda, \psi)$ with:

$$\lambda = \Lambda(x, y) \tag{2}$$

$$\psi = \Psi(x, y) \tag{3}$$





Following Lawniczak (2012), the grid distortion is described using a *Poisson* equation. The elliptic equation is broadly utilized in mechanical engineering and theoretical physics to describe how an object diffuses in space given a *charge* (Hazewinkel, 1994). The re-positioned grid locations $(\lambda, \psi)$ are the solutions of two 2D *Poisson* equations with at the right hand side the charges or *distortion functions* $\mathcal{P}$ and $\mathcal{Q}$:

$$\frac{\partial^2 \Lambda}{\partial x^2} + \frac{\partial^2 \Lambda}{\partial y^2} \quad = \quad \mathcal{P}(x,y) \tag{4}$$

$$\frac{\partial^2 \Psi}{\partial x^2} + \frac{\partial^2 \Psi}{\partial y^2} \quad = \quad \mathcal{Q}(x,y) \tag{5}$$

The distortion functions $\mathcal{P}$ and $\mathcal{Q}$ that drive the grid distortion are initially unknown, and their optimal values are to be calculated as part of the data assimilation procedure described in Section 4.3.

The second-order derivatives in Eq.(4) and (5) are discretized on the grid using finite differences. For Eq. (4), the discretization is:

$$\frac{\lambda_{i+1,j} - 2\lambda_{i,j} + \lambda_{i-1,j}}{(\Delta x)^2} + \frac{\lambda_{i,j+1} - 2\lambda_{i,j} + \lambda_{i,j-1}}{(\Delta y)^2} = P_{i,j} \tag{6}$$

and a similar discretization for Eq. (5). When this system is solved for a given right hand side, the result is a grid of 2D locations $(\lambda_{i,j}, \psi_{i,j})$ corresponding to the distorted positions of the original grid points $(x_i, y_j)$. This system can be solved using a numerical method for linear equations. In our experiments, we use the *Red-Black* ordering *Gauss-Seidel* method (Saad, 2003) to solve the discrete system of linear equations.

The distorted dust plume is interpolated back to the Cartesian grids using nearest searching method (Cayton, 2008) for comparison with observations (that are defined on longitude/latitude coordinates), and to serve as initial fields for following simulation steps.

## 4.3 Distortion estimation using 4DEnVar

The grid distortion method provides a new way for re-positioning the dust plume without adjusting the long-distance dust transport. We use the ensemble-based variational (4DEnVar) data assimilation (Liu et al., 2008) algorithm to optimize the grid distortion.

To find the optimal distortion, the initial value and covariance of $\mathcal{P}$ and $\mathcal{Q}$ need to be defined first. Each element in the two distortion equations is assumed to have a zero mean and a standard deviation, empirically chosen to be 0.015 . To enforce a smooth grid distortion, we also prescribe a correlation $c$ between two elements $\mathcal{P}(x_i, y_j)$ and $\mathcal{P}(x_k, y_l)$ (and similar for $\mathcal{Q}$):

$$c = e^{-d(x_i, y_j; x_k, y_l)/\mathcal{L}} \tag{7}$$

where $d$ represents the spatial distance in km, and $\mathcal{L}$ is an empirical length scale that is set to 1000 km. The parameters used in this study (standard deviation, correlation length scale) were chosen based on experiments for the described dust event, for other events they might need to be revised.

In the 3D model, the grid distortion is applied in horizontal direction only, changing each layer in the same way. This is mainly to reduce the degrees of freedom in the distortion, since no information on the 3D structure of the plume is available





from the observations (surface data and satellite retrieved column information). It is however also possible to use a 3D distortion with a few degrees of freedom in the vertical (Nehrkorn et al., 2015).

An ensemble of random distortion fields is generated using the assumed prior value (zero) and the assumed covariance. Each member is a vector $s$ collecting all elements of $\mathcal{P}$ and $\mathcal{Q}$ on the discrete grid:

$$[s_1, \dots, s_N] \tag{8}$$

In our experiments the ensemble size $N$ was set to 100. For each of these ensemble members, the distorted grid $(\lambda, \psi)$ is solved from the system of the discrete *Poisson* equations as described in section 4.2. With this an ensemble of distorted dust maps is formed from the *a priori* dust field $x$:

$$[x(s_1), \dots, x(s_N)] \tag{9}$$

where $x(s_i)$ represents the distorted dust field using distortion $s_i$.

Denote the ensemble *perturbation matrix* or *covariance square root* by:

$$\mathbf{S}' \; = \; \frac{1}{\sqrt{N-1}}[s_1 - s_b, \dots, s_N - s_b] \tag{10}$$

where $s_b$ is the (zero) prior value. In a 4DEnVar assimilation, the optimal distortion vector $s_a$ is defined to be a weighted sum of the columns of the perturbation matrix $\mathbf{S}'$ using weights from a control variable vector $w$:

$$s_a \; = \; s_b \; + \; \mathbf{S}'w \tag{11}$$

The optimal control variables are then calculated through minimizing of the cost function:

$$J(w) = \frac{1}{2}w^T w \; + \; \frac{1}{2}(\mathbf{HXS}'_b w \; + \; d)^T \, \mathbf{R}^{-1} \, (\mathbf{HXS}'_b w \; + \; d) \tag{12}$$

In here, $d$ is referred to as the *innovation* that describes the difference between observations $y$ and simulations on the distorted grid:

$$d \; = \; \mathbf{H}\, x(s_b) \; - \; y \tag{13}$$

In here, $\mathbf{H}$ is the *observation operator* that simulates the observed value on the distorted grid, which here simply takes the model simulation from the grid cell holding the observation location. The distortion uncertainty is transferred into the observation space through application of $\mathbf{H}$ on the ensemble members:

$$\mathbf{HXS}'_b \approx \frac{1}{\sqrt{N-1}}[\mathbf{H}x(s_1) \; - \; \mathbf{H}x(s_b), \dots, \mathbf{H}x(s_N) \; - \; \mathbf{H}x(s_b)] \tag{14}$$

The *observation representation error* $\mathbf{R}$ describes the possible differences between simulations and observations due to model and observation errors, and is here defined following Jin et al. (2018).

To ensure that the position correction is not too much influenced by differences in dust intensity, both the observations $y$ and prior dust simulations $x$ are normalized using their maximum values. Elements in $\mathbf{R}$ are also scaled using the square of the maximum observed value.





**Table 1.** Definition of assimilation experiments.

| Experiment | target error | description |
|---|---|---|
| *a priori* | - | pure model, no assimilation |
| *emis inversion* | intensity | emission inversion |
| *grid distorted assim* | position | grid distorted data assimilation |
| *hybrid assim* | position and intensity | emission inversion based on *grid distorted assim* |

## 5   Dust storm data assimilation

The grid distorted data assimilation was introduced for re-positioning the simulated dust clouds. To evaluate the effectiveness, assimilation experiments including grid distortion have been performed and compared with a traditional assimilation focusing on intensities only and a *hybrid assimilation* that combines these two. An *a priori* simulation serves as reference for all
assimilation experiments. The *emission inversion* assimilation corrects for the dust intensity errors only, while the *grid distorted assimilation* only corrects for the position error. The *hybrid assimilation* combines both in order to correct for the intensity as well as the position error.

### 5.1   Assimilation methods

Fig. 6 shows the schematic overview of the three assimilation methods listed in Table. 1. The left panel shows the setup
of the *emission inversion*, as described in detail in (Jin et al., 2019b, 2020). The inversion combines the transport model (LOTOS-EUROS) with a four dimensional variation (4DVar) data assimilation using a reduced-tangent-linearization (Jin et al., 2018). The system assumes that the processes of dust transport and removal are simulated correctly while only the emission is imperfect. The uncertainty in the emissions was parameterized as a sum of two sources: the uncertainty in the friction velocity threshold, and in the erosive wind fields. The dust emissions intensity in the source regions is then nudged to make simulation
and observations in harmony. The optimized emission fields could then be used to drive simulations that have a better forecast skill than simulations with the original emissions.

The *grid distorted assim* is designed to adjust the position of the simulated dust plume only. As described in Section. 4.3, the impact of the actual dust concentrations is avoided by normalizing the dust simulations and observations using their maximum values before calculation the distortion; afterwards, the distorted dust field is multiplied with the same maximum value again.

The right panel of Fig. 6 shows the setup of the *hybrid assim*. Different form the *emis inversion* and *grid distorted assim*, the *hybrid assim* performs two assimilations sequentially. First the *grid distorted assim* will be conducted for re-positioning the simulated dust plume. Then, the position-corrected dust plume will be used as prior in the second assimilation (similar to an *emis inversion*) to adjust the emissions to have the best possible match between actual (not normalized) observations and position-corrected simulations. The *posterior* dust field from the *hybrid assim* is then used as the initial condition for forecast
simulations.





In all assimilation tests, only observations from the snapshot of May 05 15:00 are used for fair comparison. The re-positioned plume is only available for this single moment; measurements at earlier time can therefore not be accurately assimilated in *hybrid assim*, since the corresponding simulation still has a position error then.

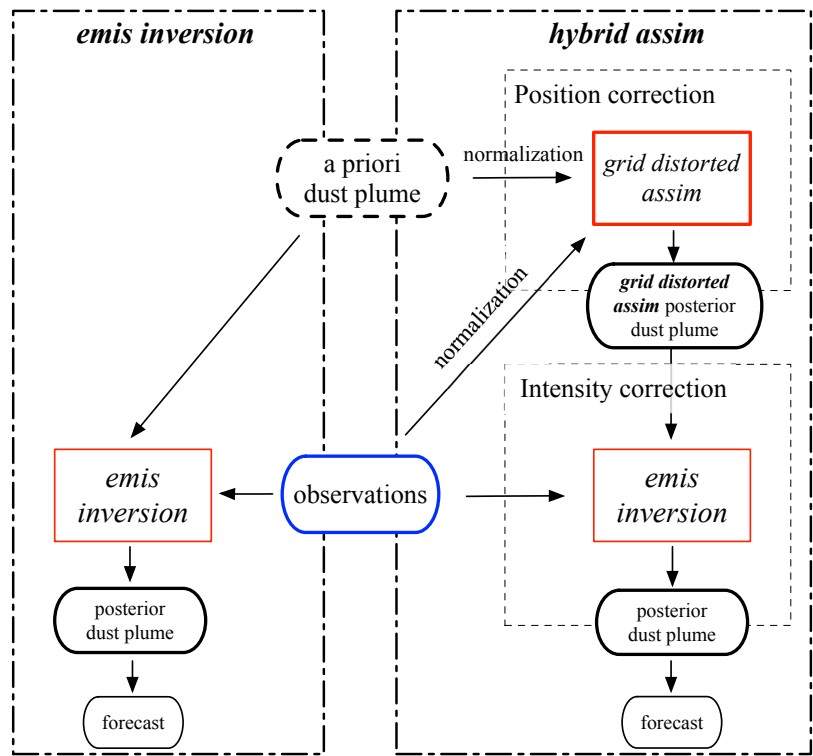

**Figure 6.** Diagrams of *emis inversion*, *grid distorted assim* and *hybrid assim* systems

## 5.2 Optimized plume position and dust load

5    The *a priori* dust plume described in Section 3.1 is assimilated with observations using the *emis inversion*, or the *grid distorted assim*, or the *hybrid assim*. The posterior surface concentrations are shown in Fig. 2 (d)~(f), respectively. The optimized dust plumes are evaluated by their position, and the RMSE metric that was introduced in Section. 3.1 to quantify the difference with observations.

Panel (d) shows the *posterior* dust plume using the *emis inversion*. The markers indicate that it has in general the same

10   position as the *a priori*, and hence the position error has not been corrected yet. In terms of root mean square error (RMSE), the *emis inversion* posterior simulation is improved but slightly; the RMSE is reduced from 388.1 $\mu$g/m$^3$ for the *a priori* simulation to 362.9 $\mu$g/m$^3$.

Using the *grid distorted assim*, the re-positioned dust plume in panel (e) matches well with the ground observations shown in panel (a.2). The marker indicating the left side of the plume is now around 35°N which is in agreement with the observations;





also the markers at the center and the right side are now better positioned. Only the very left part of the re-positioned dust plume (west of 110°E) still shows a discrepancy compared to the PM$_{10}$ observations. This can be explained from the fact that this part of the dust plume has a relatively low dust load, which makes the corresponding position error less important in the cost function Eq. 12. In addition, a rather large grid distortion is required for this part of the dust plume to match

the measurements, which is constrained with the assumed covariance of the distortion function. The RMSE of the *posterior* simulation is now significantly reduced to 251.1 $\mu$g/m$^3$. Though the dust plume is now correctly re-positioned, the simulated dust concentration does not exactly match the actual measurements. Especially in the plume center, the *posterior* simulation show dust concentrations over 1200 $\mu$g/m$^3$ that are still similar to the *a priori* simulation; while the bias-corrected observations indicate that the dust intensity in most stations are lower than 1200 $\mu$g/m$^3$.

The *hybrid assim* posterior simulation provides the best performance as shown in panel (f). The dust plume is not only re-aligned with the observations, but also the amplitude of the dust loads agrees better with the actual situation. For instance, the dust concentration in the plume center is reduced from 1500 $\mu$g/m$^3$ to 1200 $\mu$g/m$^3$, and in the upper left part of the plume the concentration level is lifted from 100 to 200 $\mu$g/m$^3$. As a result, the RMSE in the *hybrid assim* is reduced to 223.4 $\mu$g/m$^3$.

### 5.3 Forecast of dust plume position

In an operational setting the *posterior* dust concentrations are used as initial conditions for a forecast. Starting from the analysis results, forecast runs have been performed. A snapshot of the resulting forecast of the surface dust concentrations as well as the PM$_{10}$ measurements, bias-corrected dust observations, and the *a priori* forecast at 21:00 May 05 are shown in Fig. 7.

    The ground observations in panel a.1 and a.2 indicate that the dust plume is now located along 35 °N. Both in the *a priori* simulation and in the forecast based on *emis inversion*, the plume right, center and left markers are about 100, 300 and 200

20 km further south, respectively. However, the forecasts based on *grid distorted assim* or *hybrid assim* assimilation both show plumes with positions in better agreement with the observations. Best results are obtained for the *hybrid assim*, which shows better agreement for the central and upper right part of the dust field (panel f) compared to the *grid distorted* result (panel e).

### 5.4 Time series at stations

Fig.8 shows times series of dust concentrations at 6 different observation sites. The locations can be found in Fig. 1, and were

25 selected to illustrate the general results but also challenges to be solved in future. The time series show PM$_{10}$ observations (red circles), bias-corrected observations representing the dust part (red dot), the *a priori* forecast (black line), and the forecasts driven by the three assimilation tests starting from 15:00 May 05.

    For all the 6 sites, the *a priori* dust simulations estimate an arrival time of the dust cloud that is at least 4 hours too early. The *emis inversion* focusing on intensity correction does not improve the forecast of the arrival time since it only changes the

30 emission strength. Ignoring the intensity of the dust load, the temporal profiles of the dust forecasts driven by the *grid distorted assim* after May 05 15:00 are in good agreement with the temporal profile of the dust observations.

    For stations in the upper side of the plume, e.g., Baoji in panel (a), the declining trend predicted by the *a priori* and *emis inversion* forecasts are well reproduced by the *grid distorted assim*. For sites where the descend pattern was not captured by



**Figure 7.** $PM_{10}$ (a.1), bias-corrected dust observations (a.2); the *a priori* forecast (b), ensemble maximum (c) dust forecast, and dust forecast driven by *emis inversion* posterior (d); the *grid distorted assim* posterior (e) and *hybrid assim* posterior simulation (f) at 21:00 May 05. SDC: surface dust concentration.



the *a priori* simulation, the *emis inversion* helps little while *grid distorted assim* resolves the decreasing trend, as can be seen in Zibo and Zhaoyuan. For stations downwind of the plume like Xinxiang, Xuchang and Bozhou, the dust concentrations show an up and down pattern caused by the arrival and departure of the plume. The *a priori* and *emis inversion* forecasts are unable to capture the dust profile. For instance in Bouzhou, the *a priori* simulation indicated that the main dust plume arrived earlier

5  than 00:00 May 05, and it started to decline from 12:00. However, the real observation showed that the dust storm actually arrived around 12:00, with a steady increase in concentration. Starting from the *grid distorted assim*, the forecast shows concentrations with a trend similar to the observations, although the increase starts a few hours too early. The observation-minus-simulation discrepancy is further reduced for most stations using the *hybrid assim* that combines the *grid distorted assim* and *emis inversion*.

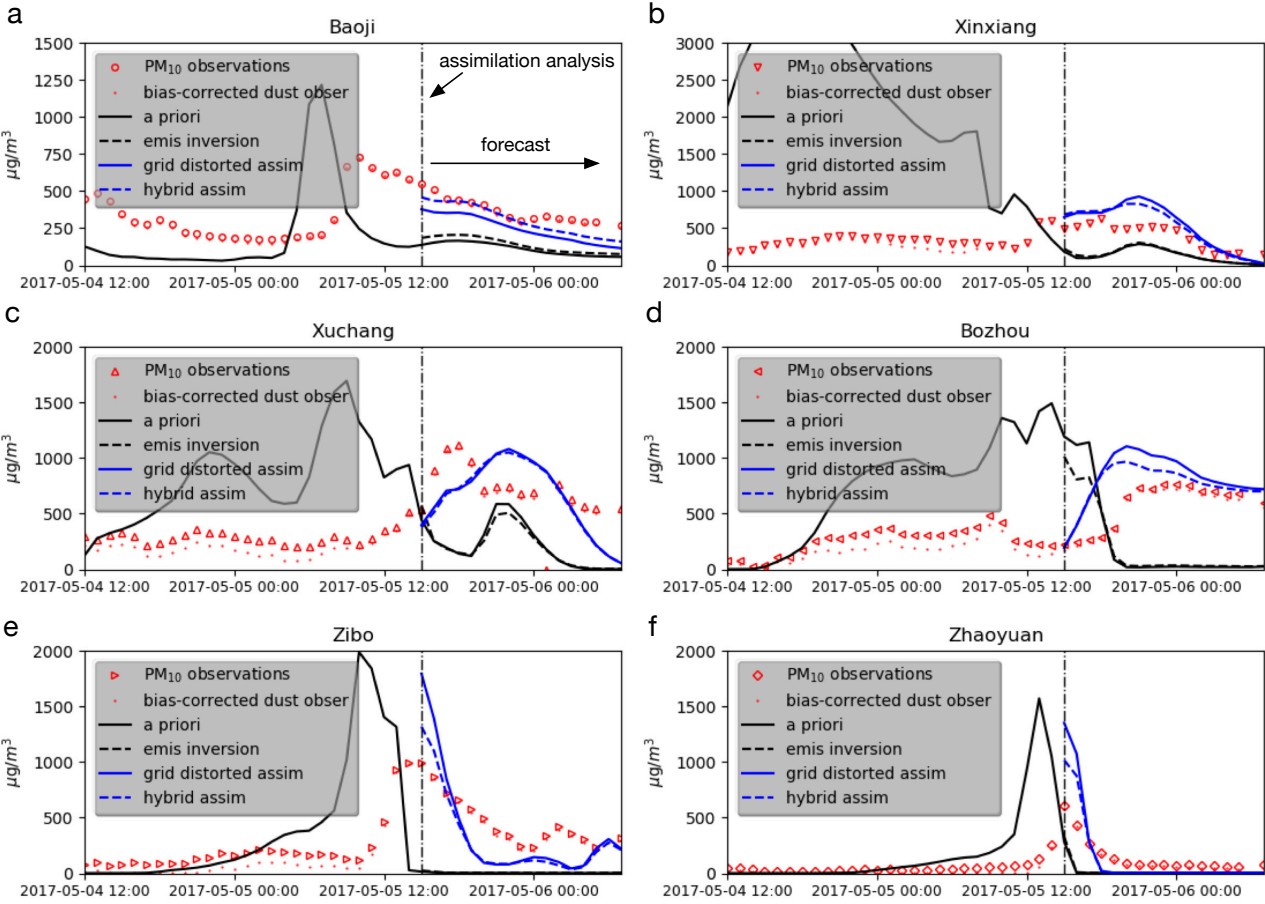

**Figure 8.** Time series of PM$_{10}$ measurements, bias-corrected dust observations, *a priori* simulation, and forecast driven by the initial state from *emis inversion*, *grid distorted assim* and *hybrid assim* at Baoji (a), Xinxiang (b), Xuchang (c), Bozhou (d), Zibo (e) and Zhaoyuan (f). The vertical black dash line indicate the start of the forecast.





## 5.5 Evaluation of forecast skills

The forecast skill of the three assimilation algorithms are also evaluated using the RMSE indicator that was also used for the *a priori* and *posterior* dust simulations in Section. 3.1 and Section. 5.2.

During the period from 16:00 May 05 to 07:00 May 06, the *a priori* RMSE reached values around 300 $\mu$g/m$^3$. The assimila-5 tion based on *emis inversion* helped to decrease the RMSE of the forecast simulations with about 20 $\mu$g/m$^3$. The improvement is limited since position errors are dominant and still present. The *grid distorted assim* is efficient in enhancing dust forecast skills in terms of the RMSE, which significantly reduce to less than 200 $\mu$g/m$^3$. When combined with *emis inversion* in the hybride approach, an additional decrease in RMSE of about 20 $\mu$g/m$^3$ is achieved.

These results show that the *grid distorted assim* is capable of correcting the position error in the simulated dust plume 10 effectively; the *hybrid assim* that combines the *grid distorted assim* and *emis inversion* provides the best initial condition to drive the dust forecast in short term.

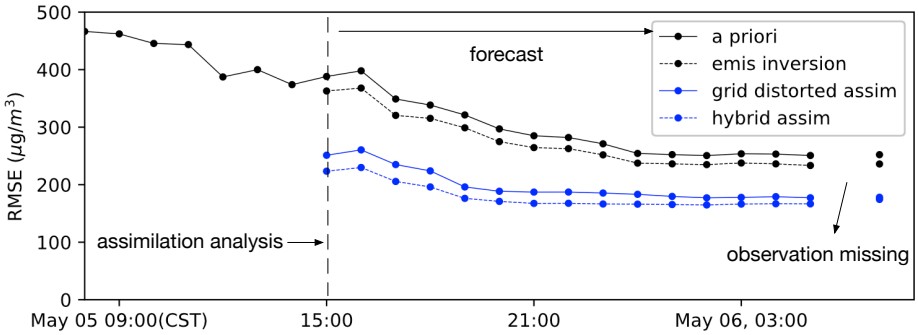

**Figure 9.** RMSE of the *a priori* dust simulation, and forecast using initial state from *emis inversion*, *grid distorted assim* and *hybrid assim*.

## 6 Summary and conclusion

Evaluation of dust storm forecasts focuses on two main criteria: the intensity of the dust load, and the position of the cloud. Various studies on improving dust forecasts focused mainly on correcting the intensity only. However, positional misfits are 15 unavoidable as a result of inaccurate emission timing profile and/or accumulation of uncertainties in long-distance transport, and therefore need to be taken into account too.

An extremely severe dust storm in May 2017 over East Asia was used as the test case in this study. A regional chemical transport model, LOTOS-EUROS, was used to reproduce the dust event. PM$_{10}$ observations are available from the China Ministry of Environmental Protection(MEP) air quality monitoring network; bias-correction was used to process the original 20 PM$_{10}$ measurements to accurately represent the dust load. The position misfits are obviously detected in the results especially when the simulated dust plume is transported thousands kilometers away to center China.



The positional misfit in dust simulation could be corrected by data assimilation too. Traditional assimilation approached require definition of a background error covariance that should account for the observation/simulation positional discrepancy. This covariance could for example include the meteorological uncertainty, as described by a meteorological ensemble forecast. For the dust storm studied here it was however shown that the spread in meteorological conditions is not sufficient to explain the position error in the simulations.

In this paper, an imaging morphing method, *grid distorted*, is adopted to re-position the simulated dust plume. The method is then combined with 4DEnVar for a *grid distorted data assimilation*, which focuses solely on correcting the dust field position to best fit the assimilated observations. Since in reality both position and intensity errors might be at present, a *hybrid assimilation* algorithm is proposed. In this hybrid system, the *grid distorted data assimilation* and a intensity-centered *emission inversion* are performed after each other.

Assimilation tests using either the *emission inversion* or *grid distorted data assimilation* only, or using the *hybrid assimilation* have been conducted on the studied dust event. The posterior dust simulation and the forecast are slightly improved by using *emission inversion*. This indicates that the traditional intensity-centered data assimilation is of little help in case positional errors are present. Only using the *grid distorted data assimilation*, strongly improved posterior and forecast simulations are obtained. The best results are obtained when the *hybrid assimilation* is performed, with both the position and intensity errors are corrected.

The *grid distorted assimilation* should be seen as an extension to traditional intensity-centered assimilation, not as a replacement. Under the presence of position error, applying grid distortion should be a pre-processing procedure to correct for errors that are not resolved otherwise.

## Code and data availability

The source code and user guidance of the CTM, Lotos-Euros, can be obtained from https://lotos-euros.tno.nl. The grid distorted data assimilation algorithm is in python environment, and is archived on Zenodo (https://doi.org/10.5281/zenodo.4579960). The real-time $PM_{10}$ data are from the network established by the China Ministry of Environmental Protection and accessible to the public at http://106.37.208.233:20035/. The observations covering the dust event is also archived on Zenodo (https://doi.org/10.5281/zenodo.4579953).

## Author contribution

JJ and AS conceived the study and designed the grid distorted data assimilation. JJ and AS performed the control and assimilation tests and carried out the data analysis. AS, HL, HXL, BH, XW, and AH provided useful comments on the paper. JJ prepared the manuscript with contributions from AS and all others co-authors.





**Competing interests**

The authors declare that they have no conflict of interest.





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
