# Peer review of "Position correction in dust storm forecast using LOTOS-EUROS v2.1: grid distorted data assimilation v1.0"

_Geoscientific Model Development, 2021_

## Author Comment (AC1)

**Response to Referee #1:** We would like to thank the referee for the careful review and the useful comments. Especially the suggestion for more detailed explanation on the cause of position error helped us to improve the quality of the manuscript.

Our response follows (*the reviewer's comments are in italics and blue*)

*General Comments:*

*The authors present the development of a novel data assimilation method, grid distorted data assimilation, to correct the position of dust plumes. They apply this technique to a case study of a dust storm event in East Asia with or without the additional correction of dust load intensity. By validating their simulation against reference observations, they show the benefit of correcting both position and intensity. The paper is clearly written and well structured. There are however some aspects that should be clarified. I have some questions for the authors and some comments that could help to improve their manuscript.*

**Specific comments:**

*1) In 5.1 when the authors describe the emission inversion, they state "The optimized emission fields could then be used to drive simulations that have a better forecast skill than simulations with the original emissions.". Hence, first they estimate the emission through data assimilation and then they use the corrected emissions to re-run the simulation? This is not what the Diagram of Fig 6 shows. Could you please clarify the use of the estimated emissions. Also, how are the emission estimated exactly? Are you taking into account that current dust concentrations are the results of emissions activated in previous times? Do you estimate emission over a sufficiently long assimilation window?*

**Reply**: Thanks for pointing out this issue. The Fig. 6 on page 15 (also shown below) is now modified, it now shows that in the ***emis inversion*** we would first generate the posterior emission field, which is then used during a 'restart' of the LOTOS-EUROS model to produce the posterior dust plume as well as the forecast. On page 14 the lines 4-6 are changed into:

"***The dust emission intensity in the source regions is then optimized such that the amplitude of the simulated concentrations is as close to the observations as possible. The optimized emission fields could then be used to drive simulations that have a better forecast skill than simulations with the original emissions***".

[Figure]

**Figure 6. Diagrams of *emis inversion*, *grid distorted assim* and *hybrid assim* systems**

The time window that is used for the inversion has been clarified on page 14, line 18-19:

"***In the* emis inversion*, the assimilation window is set from the May 02, 8:00 CST which fully covered the related dust emission for this event.*"

*2) The authors are omitting a discussion on the assimilation of vertical profiles of extinction or backscatter coefficients. By adjusting the vertical structure, their assimilation has the potential to correct the plume position.*

**Reply:** Assimilation of dust observation with vertical information would indeed further improve the 3D dust plume structure, and should be pointed out in this paper. Remarks on this have been added to page 12 line 12-16, "***In our 3D model, the grid distortion is applied in horizontal direction only, changing each layer in the same way. This is mainly to reduce the degrees of freedom in the distortion, since no information on the 3D structure of the plume is available from the current observations (surface data and satellite retrieved column information). It is however also possible to use a 3D distortion with a few degrees of freedom in the vertical (Nehrkorn et al., 2015) for dust events measurements of the vertical structure are available, e.g., lidar backscatter coefficient (Madonna et al., 2015)*.**"

The 3D dust plume structure optimization with remote sensing observations is in our future research plan, that is also mentioned in ***Conclusion*** on page 20 line 18-19, "***The method could be used to further explore 3D dust/aerosol structure by combining the 3D grid distortion and observations with vertical layering information.***"

*3) They should acknowledge that the ensemble they have used might not characterize well the uncertainty of their model. They state in the conclusions and, with similar wording also in other parts of the paper, that "For the dust storm studied here it was however shown that the spread in meteorological conditions is not sufficient to explain the position error in the simulations". They are excluding the fact that the uncertainty in the meteorology might be not well represented by that ensemble which, for example, is based on an original different horizontal (7 km) and finer vertical resolution compared their simulation (0.25 degrees and only 8 vertical layers).*

**Reply:** We agree with the reviewer that the limited ensemble meteorology data may not fully represent the uncertainty in the meteorology, either because the required uncertainty is not present there, or because of the way how our simulations make use of it. It should also be noted that the resolution of the ensemble is lower (about 30 km) than the meteorological forecast (7 km). To make it more clear, the resolution of the ensemble is now mentioned in the text on page 8 lines 12-13:

**"… to the deterministic forecast. The resolution of meteorological ensemble is about 30 km, which is comparable to the LOTOS-EUROS resolution for these experiments."**

Further, the text on page 9 lines 6-8 is changed to:

"***The uncertainty approximated using the available meteorological ensemble therefore could not be used to fully account for the position error. The origin might be that the required case is not represented in the ensemble, but also because the simulated dust transport in the LOTOS-EUROS model does not take all meteorological details into account, or is simply not accurate enough.***"

In the ***Conclusion*** on page 19-20, lines 22-2, the text is changed to

"***For the dust storm studied here it was however shown that the spread in the available meteorological ensemble, and/or the way in which the simulation model is using it, is not sufficient to explain the position error in the simulations.***".

Remarks are added on how we drive the LOTOS-EUROS using the ECMWF on page 5, line 4-6: "***An interface to the ECMWF output set is designed, which not only interpolates the default 3-hour ECMWF short-term forecast meteorology to hour values, but also averages the forecast to fit the LOTOS-EUROS spatial resolutions* (Manders et al., 2017).**" We also agree that the error from the interface needs to be taken in account while constructing the

background covariance, remarks are added on page 9, line 8-10, "*To resolve the position error, a complex covariance matrix would then be required to fully account for the accumulation of uncertainties along the long track of the plume. The uncertainty of the interface that interpolates and averages the meteorological forecast to fit our LOTOS-EUROS model resolution should also be taken into account here.*"

*4) Related to the point above, they stated in the conclusions that "Traditional assimilation approached require definition of a background error covariance that should account for the observation/simulation positional discrepancy". They should emphasize also in the rest of the paper that they are offering an alternative solution to what can be solved with what they call a "traditional assimilation approach", since the message they convey throughout the paper is that the position of the dust plume cannot be corrected by the current used assimilation algorithms. I agree that little has been done in this respect, but it would be worth investigating whether more work on the characterization of model uncertainty (either through the definition of a covariance matrix or the design of an ensemble) could account for the position discrepancy between observations and simulations.*

**Reply:** We agree that the traditional assimilation approach would solve the position error, if the background covariance matrix accounts for this correctly. The wording "approach" should be interpreted here also as "chosen configuration", since in principle every assimilation method could be able to correct for a position error, but that for practical reasons this is hardly ever done. We therefore changed fore example the abstract on page 1 line 8 to:

".. a traditional assimilation **configuration** that focuses …"

Similar in the conclusions on page 19-20, line 19-4:

**"Assimilation configurations for this type of applications usually require definition of a background error covariance or an ensemble perturbation scheme that could resolve the full observations/simulation positional discrepancy too. This covariance could for example include the meteorological uncertainty, as described by a meteorological ensemble forecast.** *For the dust storm studied here it was however shown that the spread in the available meteorological ensemble is not sufficient to explain the position error in the simulations. Therefore, a complex covariance model that could account for the accumulation of uncertainties along the long track of the plume would be required, and a substantial amount of measurements would then be necessary to constrain the transport pattern.*"

*5) I miss more explanation on the modeling scheme used: emission scheme, transport, how the ECMWF forecast drives their simulation: is it a nudging? How do they cope with the different resolution and number of vertical levels? Some of these details could point to the cause of the mismatch with the observations.*

**Reply:** The ECMWF meteorological data is averaged over the grid cells of the LOTOS-EUROS model to drive transport, emissions, and other processes. LOTOS-EUROS therefore an offline model, without computing its own meteorology. In response to Question 3, a clarification has been added.

*6) In 3.3. can you explain why the maximum and not the standard deviation of the ensemble has been used to estimate the spread in simulated dust concentrations? Could this affect your results?*

**Reply:** The maximum over the ensemble was used as a quick criterial. Only if the dust plume (in observational view) is covered by the maximum, then meteorological uncertainty (represented by ensemble meteo data) is likely to resolve the dust plume position error. While the combination of the STD and the a priori model simulation could also indicate the most likely dust plume positions, but it would miss the more extreme possibilities that eventually contain the correct case. To make this clear, explanation is added on page 8-9, line 19-2 "***The ensemble maximum here is used as a quick criterion: only if the dust plume (in observational view) is covered by the maximum, then meteorological uncertainty (represented by ensemble meteorological inputs) is likely to resolve the dust plume position error.***"

*7) In 5.1 they should add an analysis of the additional computational burden when running the hybrid assimilation system. Is it feasible for an operational forecast? Or for which application?*

**Reply:** A remark on the costs of the grid distorted data assimilation is now added on page 13, line 16-20 by saying "***The computation of the N=100 grid distortion are the most time-consuming part of the 4DEnVar based grid distorted data assimilation method, each of them costs around 2 minutes in our computing platform (CPU: Intel Xeon(R) E5; programming language: Python 3.7.6). The computation of the ensemble distortions could be re-implemented in a more efficient language, but also be easily parallelized; the grid distorted assimilation method is therefore expected to be computationally efficient enough to allow implementation in an operational forecast.***"

Also in the ***Conclusion*** on page 20, line 17-18, the follow text was added:

"***In presence of a position error, grid distorted data assimilation is a computationally efficient pre-processing procedure to correct for errors that are not resolved otherwise.***"

*8) Could you please add more information about the observation uncertainty used?*

**Reply:** Remarks are added on page 13 line 10-15:

"*The observation error covariance matrix R describes the possible differences between simulations and observations due to observation representation errors. R here is defined as a diagonal matrix, in which each representation error is set to an observation-dependent value ranging from 100 to 200 ug/m3 following Jin et al. (2018).*

*To ensure that the position correction is not too much influenced by differences in dust intensity, both the observations y and prior dust simulations x are normalized using their maximum values. Elements in R are also then scaled using the square of the maximum observed value.*"

*9) In 5.2 are you using assimilated (not independent) observations to calculate the RMSE metric? Could you please clarify this point?*

**Reply:** This is indeed important. Remarks are now added on page 14, line 24-26:

"*Note that observations that are used to evaluate the posterior performance are the same as that have been assimilated. When evaluating the method over a longer time period (multiple dust events), validation with independent observations should be considered.*"

---

## Author Comment (AC2)

**Response to Referee #2**: We would like to thank the referee for the careful review and thoughtful suggestion, which helps us to improve the quality of the manuscript.

Our response follows (*the reviewer's comments are in italics and blue*)

***General Comments:***

*Dust intensity and position errors are commonly found in model studies and difficult to be corrected especially the dust position error. In this study, the authors present a grid distorted data assimilation with the ensemble-based variational (4DEnVar) method to reduce the simulated dust position error for a dust storm event in May 2017 over East Asia. Their results demonstrate that the hybrid assimilation can correct both the dust position and intensity to provide the best initial condition for dust forecast. Generally speaking, the manuscript is scientifically sound and well-written. I recommend accepting it after addressing the following comments.*

***Major Comments:***

*According to the results, the improvements of the dust simulation and forecast are very limited with the dust emission inversion only. What is happened? Does your model successfully simulate the place and time of the dust emissions? Probably, this is due to you only show the comparisons on May 5. Can you also compare the results on start time of the dust event probably on May 2 or 3? And it is helpful to show the spatial and temporal differences of the dust emissions between the a priori and the emis inversion experiments.*

**Reply**: An incorrect emission timing profile is indeed a potential origin of the position error. However, during the first two days after the dust emission started, the simulated dust plume was still in north China and showed a good match in position with the observation. This can be seen in the Himawari-8 AOD vs. model simulation on May 3 and 4 in our previous study (Jin et al., 2019b). The good phase match in general can also be seen from a snapshot of the ground $PM_{10}$ observation vs. the simulated surface dust concentration at May 04 15:00 in Fig. 3 (also shown below). The good match in position between simulated and observed dust plume indicates that the emission timing profile is rather accurate too. When the dust plume is transported further southward, the simulated plume starts to deviate from the available surface measurements.

This has been described in ***Section 3.2 Uncertainty in emission timing profile*** on page 6, line 29-32, and page 8, line 1-7, by saying "***One potential origin of the position error is an incorrect emission time***

*profile. That is, changes in the time period over which dust is released from the source regions could to some extent alter the position of the simulated plume.*

*Actually during the first 48 hours after dust emission started, the simulated dust plume was still in north China and showed in general the same pattern as visible in the observations. For example the aerosol optical depth (AOD) retrieved from the Himawari-8 geostationary satellite showed that the simulated plumes are correctly positioned in north China (Jin et al., 2019b). The good phase match in general can also be seen from a snapshot of the ground PM$_{10}$ observation vs. the simulated surface dust concentration at May 04 15:00 (CST) in Fig. 3. There might already be position misfits in the dust simulation at these snapshots, but not easily detected. The magnitudes of the dust concentration showed discrepancies, but these could be corrected by emission inversion through assimilating those AOD observations or PM$_{10}$ measurements. The good match in position between simulated and observed dust plume indicates that the emission timing profile is rather accurate too. When the dust plume is transported further southward, the simulated plume starts to deviate from the available surface measurements.*"

[Figure]

***Figure 3. PM$_{10}$ observations (a) and the a priori dust simulation (b) at 15:00 May 04. SDC: surface dust concentration.***

A comparison between the *a priori* and emission inversion dust simulations during the early stage is added in the ***Supplementary*** (also shown below), and these show very similar values.    A remark on this is added to page 15, line 4-6, ***"The emis inversion also has little effect on the dust simulation at earlier period of the dust event, which can be found through a comparison of the a priori and emission inversion only simulations at May 03, 13:00 in Fig. S1 in Supplementary. The a priori and emis inversion also present the relative similar performance in the early stage."***

[Figure]

***Figure S1. the a priori dust simulation (a) and posterior using the emission inversion at 15:00 May 03. SDC: surface dust concentration.***

**Specific comments:**

*Please add the color bar in Figure 1.*

**Reply**: That was indeed missing. The modified Fig. 1 can be found on page 5.

*Page 6 Line 6, Alex desert -> Alxa desert?*

**Reply**: Corrected.

*Page 13 Line 20 form from?*

**Reply**: Corrected.